# Ulnar finger posture effect on a pinch strength

Shota Date[1]*, Hiroshi Kurumadani[1], Nanako Nakamae[1], Yosuke Ishii[2],
Kazuya Kurauchi[1], Naoki Maniwa[1], Naoya Goto[1], Toshiyuki Fukushima[1], Ken Hirao[3],
Toru Sunagawa[1]

1 Laboratory of Analysis and Control of Upper Extremity Function, Graduate School of Biomedical & Health Sciences, Hiroshima University, Hiroshima, Japan, 2 Department of Biomechanics, Graduate School of Biomedical & Health Sciences, Hiroshima University, Hiroshima, Japan, 3 Department of Orthopedic Surgery, Hirao Clinic, Hiroshima, Japan

* sdate@hiroshima-u.ac.jp

## Abstract

Upper extremity position affects maximum pinch strength; however, there is no consensus regarding the posture of the three ulnar fingers during pinch strength testing, and existing studies have different flexion or extension posture, depending on the measurement method used. This study aimed to clarify the effect of ulnar finger posture on maximum pinch strength and to verify its reliability. Thirty-three participants (19 males and 14 females) performed maximum pulp-to-pulp pinching with both dominant and non-dominant hands. Their ulnar finger posture was tested in two positions (flexion and extension). Measurements were conducted on three separate days by two examiners. Pinch strengths were compared for ulnar finger posture, hand dominance, and sex. The intraclass correlation coefficient was calculated to assess intra-rater, inter-rater and test-retest reliability. Pinch strength in the flexion posture was significantly stronger than that in the extension posture, regardless of hand dominance or sex (1.34 times greater). Intra-rater reliability was good to excellent in the flexion posture (0.89–0.92) and moderate to good in the extension posture (0.71–0.89). Inter-rater reliability was moderate for both flexion and extension postures (flexion: 0.64–0.68, extension: 0.52–0.54). Test-retest reliability was good to excellent in the flexion posture (0.83–0.91) and moderate in the extension posture (0.71–0.72). These results indicate the importance of hand therapists instructing patients to flex their ulnar fingers during testing to obtain maximum and reliable pinch strength measurements, thereby enhancing the accuracy of hand functional assessment in clinical practice.

## Introduction

Pinch strength, a component of finger motor performance, serves as a foundation for activity and participation [1,2]. Pinch strength is also widely used in clinical practice to assess the degree of upper extremity dysfunction [3–6]. Therefore, pinch strength

**Data availability statement:** All relevant data are within the manuscript and its Supporting Information files.

**Funding:** The authors disclose the receipt of the following financial support for the research, authorship, and/or publication of this article: JSPS KAKENHI, Grant Number JP 20K19344 and 24K205300A supported this work.

**Competing interests:** The authors have declared that no competing interests exist.

measurement is an important objective indicator of treatment progress and for hand surgeons and therapists to establish appropriate goals for patients with hand injuries [6,7].

The American Society of Hand Therapists (ASHT) recommends a standardized pinch testing position for the upper extremities, such as the shoulder, elbow, and forearm, during pinch strength measurement [1], and various studies have examined the effect of these joint positions on pinch strength [8–11], The ASHT also addresses the posture of the other fingers that are not directly involved in the pinching movement [1]. The ulnar fingers, comprising the middle, ring, and little fingers, should be flexed in the palm when performing pinching with the thumb and index finger. However, the effect of ulnar finger posture on pinching performance is not well explored [12–14], and depending on the measurement method used, the posture of the ulnar fingers differs in flexion or extension [7,15–19]. Considering the biomechanical findings that the agonist (e.g., digitorum superficialis) and antagonist muscles (e.g., extensor digitorum) involved in index finger flexion share a muscle belly with the four fingers, ulnar finger posture would affect the strength of the maximum pulp pinching between the thumb and index finger [20–23]. Despite the close association between the position of the upper extremity and motor function, there is currently no consensus among clinicians or researchers regarding ulnar finger posture during pinch strength testing. Therefore, thoroughly investigating the effect of ulnar finger posture on pinch strength to enhance the accuracy of hand functional assessment is imperative. We aimed to clarify the effect of ulnar finger posture on maximum pinch strength and verify its reliability.

## Materials and methods

### Participants

A priori power analysis was performed using the G*Power statistical packages (G*Power Ver. 3.1.9.2, Universität Düsseldorf, Düsseldorf, Germany) [24] to determine the appropriate sample size, which was determined to be 30 participants (medium effect size = 0.25; α = 0.05; power = 0.90) [25]. In total, 33 healthy volunteers (19 males and 14 females) with a mean age of 34.0 ± 13.6 years were recruited. The demographic data of the enrolled participants are listed in Table 1. None of the participants had musculoskeletal or neurological dysfunction or limitations in the range of motion of their fingers and wrists. Written informed consent was obtained from all participants before their participation. This study was approved by our institutional ethics review board (Ethics Committee for Epidemiology of Hiroshima University, approval number: E2018-1517) and conducted in accordance with the Declaration of Helsinki.

### Pulp-to-pulp pinch task

The participants performed a pulp-to-pulp pinch task using the thumb and index finger of both dominant and non-dominant hands. In accordance with the ASHT recommendation [1] and previous studies [7,28–31], participants performed the task while seated on a chair, with 0° adduction, 0° flexion, and neutral rotation of the

**Table 1. Participants' demographic data.**

| Measurement | Male (n = 19) | Female (n = 14) | Total (n = 33) | Range |
|---|---|---|---|---|
| Age (years old) | 35.3 ± 13.3 | 32.3 ± 14.4 | 34.0 ± 13.6 | [21–71] |
| Height (cm) | 172.2 ± 5.8 | 159.4 ± 4.1 | 166.8 ± 8.2 | [150.0–184.0] |
| Body mass (kg) | 68.9 ± 5.6 | 49.3 ± 5.2 | 60.4 ± 11.1 | [42.0–80.0] |
| Hand length (cm) | 18.4 ± 0.6 | 16.8 ± 0.5 | 17.7 ± 1.0 | [16.0–19.0] |
| Hand width (cm) | 8.5 ± 0.5 | 7.6 ± 0.9 | 8.1 ± 0.8 | [7.0–10.5] |
| Hand span (cm) | 20.2 ± 0.8 | 18.2 ± 1.3 | 19.3 ± 1.4 | [16.0–21.0] |
| Hand circumference (cm) | 17.7 ± 2.7 | 16.2 ± 1.9 | 18.5 ± 2.5 | [16.0–20.5] |

Data are presented as the mean ± standard deviation (SD).

Definition of the hand measurement data: hand length, the distance from the middle of inter stylion to the tip of middle finger; hand span, the distance from the tip of the thumb to the tip of the little finger with the hand opened as wide as possible; hand circumference, the superficial distance around the edge of metacarpal [26,27].

shoulder; 90° flexion of the elbow; a neutral position of the forearm; and neutral and slight dorsiflexion (0–30°) of the wrist. The thumb and index finger were slightly flexed at the metacarpophalangeal and interphalangeal joints (Fig 1A). Two ulnar finger postures were used, namely flexion and extension. The metacarpophalangeal and interphalangeal joints of the ulnar fingers were fully flexed in the flexion posture and were naturally extended in the extension posture. Regarding the extension posture, the participants were not instructed on the degree of extension of the joints to simulate a more clinical and practical pinch testing situation (Fig 1B). Three trials were performed for each ulnar finger posture with a rest period of at least 1 min between the measurements. The duration of each pinch exertion was approximately 3 s. The examiner verbally encouraged the participants to squeeze as hard as possible throughout the measurement. To prevent any influence on subsequent measurements, feedback on the measurement value was withheld from the participants. Pinch strength was measured using a digital pinch meter with a pinch width of 7 mm (Mobie; Sakai Medical Co., Ltd., Tokyo, Japan). The experimental data were recorded in kilograms of force (kgf) and stored in a database using Microsoft Excel (Microsoft Co., Redmond, WA, USA) for further analysis (see section 2.5).

## Examiners

To verify inter-rater (between-rater) reliability, two examiners (S. D. and N. N.) conducted the measurements. Examiner 1 (S. D.) was a 35-year-old male occupational therapist with 12 years of experience and clinical expertise in pinch strength testing. Examiner 2 (N. N.) was a 23-year-old female occupational therapist with two years of experience who lacked clinical practice in pinch strength testing. Examiner 2 received 30 min of measurement training from Examiner 1 to familiarize her with the equipment and experimental protocol before conducting the measurement.

## Experimental protocol

The participants performed the task with each ulnar finger posture using both dominant and non-dominant hands. The examiners did not specify the order of the flexion and extension postures, which led the participants to initially perform the pinching movement with a natural ulnar finger posture. Subsequently, the participants performed the task in other postures. The order of the hand dominance measurements (dominant or non-dominant) was random and counterbalanced. Measurements were conducted over the course of three separate days to investigate data reliability. Examiner 1 took measurements on days 1 and 2, and Examiner 2 took measurements on day 3. Each measurement day was at least two weeks apart.

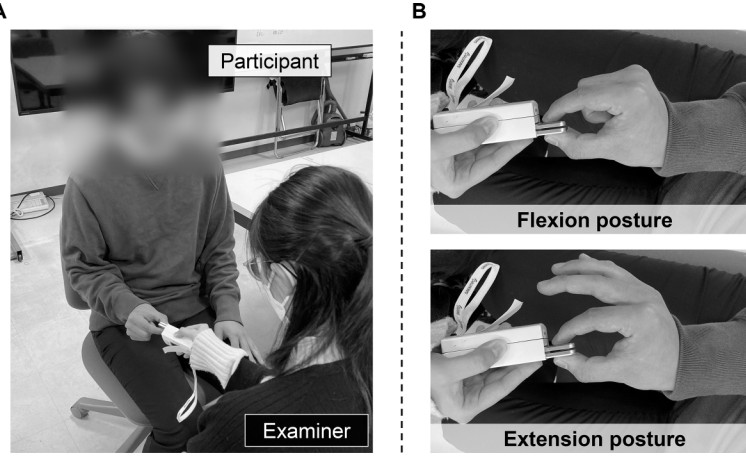

**Fig 1. Task performed by the participants.** (A) Measurement position of the participant during a pinch test. (B) The upper and lower panels represent the ulnar finger postures in the flexion and extension conditions, respectively.

## Data processing

All data were analyzed using Microsoft Excel. The proportion of postures during the initial pinching movement was calculated to determine the natural ulnar finger posture of each participant. Pinch strength was calculated as the average of three trials for each hand (dominant and non-dominant) and each ulnar finger posture (flexion and extension).

## Statistical analysis

SPSS version 29.0.2 statistical software (IBM Inc., Chicago, IL, USA) was used for statistical analysis. Pinch strengths were compared using a three-way analysis of variance (ANOVA) for ulnar finger posture (flexion and extension), hand dominance (dominant and non-dominant), and sex (male and female). Post hoc tests were performed using the Bonferroni test. The pinch strength results are presented as the mean ± standard deviation (SD). As a subgroup analysis to investigate whether pinch strength in the ulnar finger position changes based on participant's demographic data, the relationship between pinch strength ratio between the flexion and extension postures (flexion relative to extension) and age, height, body mass, and hand size was examined using a Pearson's correlation coefficient. Furthermore, the pinch strength ratio between male and female was compared using a Student's t-test.

To assess intra- and inter-rater and test-retest reliability, intraclass correlation coefficient (ICC) was analyzed with a 95% confidence interval (CI). An ICC value > 0.90 indicated excellent, 0.75–0.90 indicated good, 0.50–0.75 indicated moderate, and < 0.50 indicated poor reliability. Additionally, the standard error of measurement (SEM) and minimal detectable change with 95% confidence ($MDC_{95\%}$) were analyzed. The SEM was calculated as $SD \times \sqrt{(1-ICC)}$ and was interpreted as an estimate of the variation in measurements that would result from the true value of the participants. The $MDC_{95\%}$ was calculated as $SEM \times 1.96 \times \sqrt{2}$ and was interpreted as the smallest change in a measure that can be considered real change beyond measurement error with 95% confidence. Both the SEM and MDC are presented in the same unit of measurement as the variable itself (kgf), facilitating the ease of interpretation of the measurement error [32]. Additionally, to examine the presence or absence of systematic errors in the differences between the intra- and inter-rater and test-retest measurements, additive and proportional errors were evaluated [33–36]. Additive error was evaluated by calculating the limit of agreement (LoA) using the 95% CI of the mean of the difference between the

two measurements [34]. We considered the additive error to be absent when the LoA had intervals that included zero. Proportional error was evaluated by calculating a regression coefficient (r) of the relationship between the two measurements [36]. We considered the proportional error to be absent when the correlation was not statistically significant [33–35]. Significance level was set at < 5%.

## Results

### Proportion of the natural ulnar finger posture

The proportion of natural ulnar finger posture during the pinching movement was 66.7% (22/33) in flexion and 33.3% (11/33) in extension. Pinch strength between participants with the natural flexion and extension ulnar finger postures did not differ significantly (S1 Table).

### Maximum pinch strength

The pinch strength values for each sex, hand dominance, and ulnar finger posture are listed in Table 2. Pinch strength was a significant main factor associated with ulnar finger posture (F = 33.5, p < 0.001), hand dominance (F = 4.2, p = 0.048), and sex (F = 10.8, p < 0.001); however, there was no interaction between any factors. In the post hoc test, the pinch strength in the flexion posture was significantly stronger than that in the extension posture, and the pinch strength in the flexion posture was 1.34 times greater than that in the extension posture. Subgroup analysis indicated that the pinch strength ratio between the flexion and extension postures did not statistically vary by age (p = 0.779), height (p = 0.398), body mass (p = 0.68), hand sizes (p = 0.186–0.945), or sex (p = 0.580) (S2 Table, S3 Table). Additionally, the pinch strength of the dominant hand was 1.10 times greater than that of the non-dominant hand, and was also 1.40 times greater in males than in females.

### Measurement reliability

An overview of the ICC, 95% CI, SEM, $MDC_{95\%}$, LoA, and correlation coefficient (r) for intra- and inter-rater and test-retest reliability is listed in Table 3. Detailed values of pinch strength for each trial, examiner, and test-retest measurement are provided in S4, S5, and S6 Tables. Good to excellent intra-rater reliability was observed in the flexion posture (ICC: dominant, 0.92; non-dominant, 0.89), whereas moderate to good intra-rater reliability was observed in the extension posture (ICC: dominant, 0.89; non-dominant, 0.71). Moderate inter-rater reliability was observed for both flexion and extension postures (ICC: 0.52–0.68). Good to excellent test-retest reliability was observed in the flexion posture (ICC: dominant, 0.83; non-dominant, 0.91), whereas moderate test-retest reliability was observed in the extension posture (ICC: dominant, 0.72; non-dominant, 0.71). All the LoA had intervals that included zero, and none of the correlation coefficients (r values) were statistically significant.

**Table 2. Result of pinch strength for three-way analysis of variance (ANOVA).**

| Factor | Level | Mean ± SD | P-value |
|---|---|---|---|
| Ulnar finger posture | Flexion | 4.5 ± 1.7 | P < 0.001 |
| | Extension | 3.2 ± 1.0 | |
| Hand dominance | Dominant | 4.1 ± 1.7 | P = 0.048 |
| | Non-dominant | 3.7 ± 1.4 | |
| Sex | Male | 4.5 ± 1.7 | P < 0.001 |
| | Female | 3.2 ± 1.0 | |

Data are presented as the mean ± standard deviation (SD). Unit of the pinch strength is kilograms of force (kgf).

**Table 3. Results of intra- and inter-rater and test-retest reliability.**

| | Ulnar finger posture | Hand dominance | ICC | 95% CI | SEM (kgf) | MDC$_{95\%}$ (kgf) | LoA (kgf) | r (p) |
|---|---|---|---|---|---|---|---|---|
| Intra-rater reliability | Flexion | Dominant | 0.92 | [0.86–0.95] | 0.92 | 1.52 | [−1.42–1.85] | 0.09 (0.617) |
| | | Non-dominant | 0.89 | [0.82–0.94] | 0.89 | 1.37 | [−1.40–1.57] | 0.11 (0.534) |
| | Extension | Dominant | 0.89 | [0.82–0.94] | 0.89 | 1.35 | [−1.21–1.66] | 0.11 (0.543) |
| | | Non-dominant | 0.71 | [0.37–0.89] | 0.71 | 1.62 | [−1.64–2.29] | 0.70 (0.069) |
| Inter-rater reliability | Flexion | Dominant | 0.64 | [0.24–0.86] | 1.01 | 2.79 | [−1.43–3.17] | 0.11 (0.720) |
| | | Non-dominant | 0.68 | [0.31–0.87] | 0.88 | 2.43 | [−0.91–2.69] | 0.25 (0.330) |
| | Extension | Dominant | 0.54 | [0.76–0.97] | 0.69 | 1.90 | [−0.79–2.11] | 0.38 (0.152) |
| | | Non-dominant | 0.52 | [0.07–0.80] | 0.61 | 1.70 | [−1.23–1.97] | 0.18 (0.458) |
| Test-retest reliability | Flexion | Dominant | 0.83 | [0.58–0.93] | 0.83 | 2.12 | [−1.9–2.4] | 0.18 (0.505) |
| | | Non-dominant | 0.91 | [0.76–0.97] | 0.91 | 1.51 | [−1.6–1.6] | 0.49 (0.055) |
| | Extension | Dominant | 0.72 | [0.38–0.89] | 0.72 | 1.33 | [−1.1–1.5] | 0.12 (0.669) |
| | | Non-dominant | 0.71 | [0.37–0.89] | 0.71 | 1.21 | [−1.3–1.2] | 0.43 (0.096) |

CI = confidence interval; ICC = intraclass correlation coefficient; LoA = limit of agreement; MDC$_{95\%}$ = minimal detectable change with 95% confidence; r (p) = correlation coefficient (p-value); SEM = standard error of measurement.

## Discussion

The findings of this study revealed that pinch strength was stronger when the ulnar finger posture was flexed than when it was extended, regardless of hand dominance or sex. The measurement reliability during both the flexion and extension postures was acceptable; specifically, the ICC values during the flexion posture showed good to excellent measurement reliability.

The proportion of the natural ulnar finger posture was 33% during extension. This result suggests that in clinical practice, some patients may have unconsciously extended their ulnar fingers during a pinching test if the examiner fails to specify the ulnar finger posture. This behavior can lead to a lower value than the true value.

The pinch strength during the ulnar finger flexion posture was significantly stronger than that during the extension posture, consistent with findings from a previous study [12], which could be attributed to biomechanical factors on the agonist/antagonist muscles involved in the index finger. For flexion of the index finger during the pinching movement, the primary agonist muscles were the flexor digitorum superficialis and flexor digitorum profundus, while the antagonist muscle was the extensor digitorum [20–22]. Each muscle shares a muscle belly with the four fingers. Ulnar finger extension involuntarily exerts force in the direction of extension of the index finger [23], impeding its flexion function. Consequently, the pinch strength in the extension posture could have been weaker than that in the flexion posture [12]. In contrast, the pinch strength in the flexion posture relative to the extension posture was 1.3 times greater in the present study, differing from the previous study, where it was up to 1.9 times greater [12]. This difference could involve the instructions in the extension condition. In the previous study, the participants fully extended their ulnar fingers consciously, whereas in the current study, they were instructed to allow their ulnar fingers to extend naturally. As pinch strength testing is rarely performed with the ulnar fingers fully extended, our results regarding the ratio of pinch strength between the flexion and extension postures are more clinical and practical than those of the previous study. The pinch strength of the dominant hand was significantly stronger than that of the non-dominant hand. This result was consistent with findings from previous studies, suggesting that the difference in hand dominance could be due to the greater muscle mass in the dominant arm than in the non-dominant arm [37]. In contrast, other studies have reported no difference between the dominant and non-dominant hands in pinch strength. In the current study, the ratio of the pinch strength in the dominant hand relative to the non-dominant hand was only 1.10 times greater, thus, this dominant hand difference may be due to the methodology used across studies. The pinch strength was significantly stronger in males than in females. This sex difference was consistent

with the findings of previous studies and could be attributable to the fact that males have more muscle mass than females [38], shorter half-relaxation time, greater proportional area of Type II (fast twitch) fibers [39], and larger hand sizes.

According to the measurement reliability, intra-rater reliability ranged from moderate to excellent, while inter-rater reliability was moderate, and test-retest reliability fell within moderate to excellent range. Additionally, all the LoA had intervals that included zero, indicating the absence of additive error, and the correlations coefficients (r values) were not significant, suggesting the absence of proportional error. These results mean that the systematic error was within acceptable limits. These findings demonstrate that the pinch strength in both flexion and extension postures of the ulnar fingers remains consistent regardless of the individual conducting the test, number of repetitions, or date of assessment. Notably, the ICC values for hand dominance were higher in the flexion posture than in the extension posture. This finding suggests that pinch strength can be measured more reliably in the ulnar finger flexion posture than in the extension posture. Fully extending the ulnar fingers enhances the stability of the index finger joint during a pulp-to-pulp pinch task using the thumb and index finger [13,40]. However, this finding is inconsistent with the results of this study. This discrepancy could be attributed to the variation in the degree of ulnar finger extension during measurement of the extension condition. Therefore, maintaining the ulnar finger in a flexed position would enable reliable measurements. On the other hand, the ICC values for inter-rater reliability (0.52–0.68) were lower than those for intra-rater (0.71–0.92) and test-retest (0.71–0.91) reliability. These lower ICC values for inter-rater reliability could be attributed to the difference in experience between Examiner 1 and Examiner 2, or to the insufficient measurement training time (30 min) for Examiner 2. It appears necessary for inexperienced examiners to receive more comprehensive measurement training, such as sessions longer than 30 min or instruction by a supervisor following the ASHT recommended measurement methods [1].

Our findings support the results of a previous study on the effect of ulnar finger posture during the pinching movement [12] and could be integrated into standard protocol of the ASSH's recommended measurement methods [1]. Based on our findings, clinicians should consistently instruct patients to flex their ulnar fingers during pinch testing as the flexion posture generates greater pinch strength and exhibits less inter-, intra-rater, and test-retest variability than the extension posture. Furthermore, given the moderate ICC for the inter-rater reliability, having the same examiner perform pinch testing when longitudinally assessing a patient's pinch strength is advisable. In addition, when assessing longitudinally, a change in pinch strength of more than 2 kg can be interpreted as an improvement, consistent with a previous study [41], because $MDC_{95\%}$ represents true change beyond measurement error and the $MDC_{95\%}$ results for test-retest reliability of flexion posture in this study was around 2 kg (2.12 kg for the dominant hand, 1.51 kg for the non-dominant hand).

This study had some limitations. First, the small sample size restricted the generalizability of our results. Although our study clarified the effect of ulnar finger posture on pinch strength, recruiting a larger number of participants is necessary to standardize the pinch strength in both the flexion and extension postures of the ulnar fingers. Second, we only measured pulp-to-pulp pinching movements using the thumb and index finger. However, tripod pinching using the thumb, index, and middle fingers may also be influenced by the posture of the ring and little fingers. Future studies should compare the strength of the tripod pinch between flexion and extension of the ulnar fingers while ensuring experimental settings that minimize confounding factors such as examiner variability and fatigue. Fourth, because we did not perform biomechanical and physiological measurements using motion analysis and electromyography (EMG), whether finger joint angles during the task or how ulnar finger postures were affected by the muscle activity involved in the pinching movement remains unknown. A detailed investigation using movement analysis and EMG would be required to clarify the exact influence of ulnar finger posture during pinching movements. These biomechanical and physiological measurements could also provide insight into why the ulnar finger is extended during pinching movements. Finally, the pinch width was fixed at 7 mm due to equipment constraints, and the width could not be adjusted for each individual. Different pinch widths can increase or decrease pinch strength [8,42–45]. In the extension posture used in this study, a larger pinch width may result in greater pinch strength because involuntary extension force applied to the index finger is weakened with increasing pinch width.

## Conclusions

Significantly stronger strength during pulp-to-pulp pinching was observed in healthy participants when the ulnar finger posture was flexed than when it was extended (1.3 times greater). Measurement reliability was also higher in flexion than in extension. These findings suggest that clinicians should instruct patients to flex their ulnar fingers during testing to obtain maximum and reliable pinch strength measurements.

## Supporting information

**S1 Table. Result of pinch strength in the natural ulnar finger posture.**
(DOCX)

**S2 Table. Correlation of pinch strength ratio (flexion/ extension) with participant's demographic data.**
(DOCX)

**S3 Table. Result of pinch strength ratio (flexion/ extension) for each sex.**
(DOCX)

**S4 Table. Result of pinch strength for each trial.**
(DOCX)

**S5 Table. Result of pinch strength for each examiner.**
(DOCX)

**S6 Table. Result of pinch strength in test and retest.**
(DOCX)

**S1 Data. dataset.csv**
(CSV)

## Acknowledgments

The authors would like to thank Editage (www.editage.com) for English language editing.

## Author contributions

**Conceptualization:** Shota Date, Hiroshi Kurumadani, Toru Sunagawa.

**Data curation:** Shota Date, Nanako Nakamae.

**Formal analysis:** Shota Date.

**Funding acquisition:** Shota Date.

**Investigation:** Shota Date, Nanako Nakamae.

**Methodology:** Shota Date, Hiroshi Kurumadani, Yosuke Ishii, Toru Sunagawa.

**Project administration:** Shota Date.

**Resources:** Shota Date, Nanako Nakamae, Kazuya Kurauchi, Ken Hirao, Toru Sunagawa.

**Software:** Shota Date.

**Supervision:** Toru Sunagawa.

**Validation:** Hiroshi Kurumadani, Nanako Nakamae, Kazuya Kurauchi, Naoki Maniwa, Naoya Goto, Toshiyuki Fukushima, Toru Sunagawa.

**Visualization:** Shota Date, Nanako Nakamae, Naoki Maniwa.

**Writing – original draft:** Shota Date.

**Writing – review & editing:** Hiroshi Kurumadani, Kazuya Kurauchi, Naoya Goto, Toshiyuki Fukushima, Toru Sunagawa.

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
