## [Decision Letter · Decision Letter 0]

11 Mar 2025

Please submit your revised manuscript by Apr 25 2025 11:59PM. If you will need more time than this to complete your revisions, please reply to this message or contact the journal office at plosone@plos.org . A rebuttal letter that responds to each point raised by the academic editor and reviewer(s). You should upload this letter as a separate file labeled 'Response to Reviewers'.A marked-up copy of your manuscript that highlights changes made to the original version. You should upload this as a separate file labeled 'Revised Manuscript with Track Changes'.An unmarked version of your revised paper without tracked changes. You should upload this as a separate file labeled 'Manuscript'.

We look forward to receiving your revised manuscript.

Kind regards,

Saumya Srivastava, Ph.D.

Academic Editor

PLOS ONE

Journal Requirements:

 “The authors disclose the receipt of the following financial support for the research, authorship, and/or publication of this article: JSPS KAKENHI, Grant Number JP 20K19344 and 24K205300A supported this work.”

4. We note that your Data Availability Statement is currently as follows: “All relevant data are within the manuscript and in Supporting Information files.”

Additional Editor comments: 

Subject: Manuscript Revision – Minor Corrections

I hope this email finds you well.

The review process for your manuscript titled "Ulnar finger posture effect on a pinch strength" (Manuscript ID: [PONE-D-24-49287]) has been completed. The reviewers have suggested minor revisions to enhance the clarity and quality of your work.

Please find the reviewer's comments for your reference. Kindly revise the manuscript accordingly and submit the revised version along with a point-by-point response to the comments at your earliest convenience.

Should you have any questions or require further clarification, please do not hesitate to reach out.

Best regards,

Dr Saumya Srivastava, PhD

Academic Editor- PLOS One

We are ready with the comments received by the reviewers. The manuscript is well-structured, with **robust data and meaningful clinical implications** . With minor improvements in **methodological clarity, statistical justification, and discussion depth** , it is **suitable for acceptance**  after revision.

Reviewers' comments:

Reviewer's Responses to Questions

**Comments to the Author**

1. Is the manuscript technically sound, and do the data support the conclusions?

Reviewer #1: Yes

Reviewer #2: Yes

2. Has the statistical analysis been performed appropriately and rigorously?

Reviewer #1: Yes

Reviewer #2: Yes

3. Have the authors made all data underlying the findings in their manuscript fully available?

Reviewer #1: Yes

Reviewer #2: Yes

4. Is the manuscript presented in an intelligible fashion and written in standard English?

Reviewer #1: Yes

Reviewer #2: Yes

Reviewer #1: Thank you for the opportunity to review this interesting paper on finger postures during pinching. The experimental objectives, methods, and postoperative data evaluation methods including inter-rater evaluations are all well-documented. There are some questions regarding the discussion of the results. Please refer to the points below.

Line 212, page16; The ICC values for inter-rater reliability (0.52-0.68) are not particularly high; please include a discussion addressing this point. Could this be due to a lack of experience on the part of Examiner 2?

What are the causes of the differences in the natural ulnar finger posture during pinching? If pinch strength increases when the ring and little fingers are flexed posture, why do some individuals extend their ring and little fingers instead under power pulp pinch?

Reviewer #2: General Comments

This manuscript is an important finding for hand therapy and outcome measurement assessment of the hand. I believe this work adds to the body of literature with new knowledge but it is not without its limitations. Please see my comments below.

Can you elaborate on why you chose to focus solely on pulp-to-pulp pinch strength rather than including other pinch types (e.g., lateral or tripod pinch) in this study? How might the inclusion of other pinch types alter the findings?

The abstract mentions no consensus on ulnar finger posture during pinch strength testing. Could you provide more context on the range of practices currently observed in clinical settings beyond the ASHT recommendation?

How do you envision these findings being integrated into standardized clinical protocols for pinch strength assessment, given the variability in current practices?

You note that the effect of ulnar finger posture on pinch strength is not well explored. Are there specific biomechanical or physiological findings/issues from prior studies that guided your research question?

Participants were instructed to extend their ulnar fingers "naturally" in the extension posture. How was "natural" extension defined or ensured during testing, and did this vary significantly between participants?

The sample size was determined using a power analysis with an effect size of 0.25. Can you explain how this effect size was chosen, and whether it aligns with effect sizes reported in prior pinch strength studies?

Examiner 2 had less experience and received 30 minutes of training. Did you assess whether this training was sufficient to minimize systematic differences between examiners, beyond what the inter-rater reliability results show?

Pinch strength was measured with a 7mm pinch width. Why this specific width? How might varying pinch widths affect the observed differences between flexion and extension postures?

Pinch strength in the flexion posture was 1.34 times greater than in the extension posture. Did you explore whether this ratio varied significantly across individuals or subgroups (e.g., by age, hand size, or sex)?

Also, the natural ulnar finger posture was flexion in 66.7% of participants. Did you analyze whether participants with a natural extension posture exhibited different pinch strength patterns compared to those with a natural flexion posture?

Inter-rater reliability was moderate (ICC: 0.52–0.68), lower than intra-rater and test-retest reliability. What factors do you think contributed to this, and how might they be addressed in future studies?

The SEM and MDC95% values were provided for reliability. How do these values translate to clinically meaningful differences in pinch strength, for patient populations with hand impairments for example.

**Do you want your identity to be public for this peer review?** For information about this choice, including consent withdrawal, please see our Privacy Policy

Reviewer #1: No

Reviewer #2: **Yes: ** Gary Guerra

---

## [Author Response · Author response to Decision Letter 1]

9 Apr 2025

Journal Requirements:

RESPONSE: We prepared resubmitted manuscripts using the PLOS ONE style templates and ensured that all submissions met the PLOS ONE style requirements, including those for file naming.

RESPONSE: We have provided incorrect grant numbers in the ‘Funding Information’ section. We have revised it accordingly (JP20K19344 and JP24K20530).

3. Thank you for stating the following financial disclosure: “The authors disclose the receipt of the following financial support for the research, authorship, and/or publication of this article: JSPS KAKENHI, Grant Number JP 20K19344 and 24K205300A supported this work.” Please state what role the funders took in the study. If the funders had no role, please state: "The funders had no role in study design, data collection and analysis, decision to publish, or preparation of the manuscript." If this statement is not correct you must amend it as needed. Please include this amended Role of Funder statement in your cover letter; we will change the online submission form on your behalf.

RESPONSE: We have revised the grant numbers (JP20K19344 and JP24K20530) and specified that the funders had no role in study design, data collection and analysis, decision to publish, or preparation in our cover letter as follows:

“This work was supported by Japan Society for Promotion of Science (JSPS) KAKENHI (Grant-in-Aid for Early-Career Scientists) awarded to S.D. [grant numbers JP20K19344 and JP24K20530]. JSPS website: [https://www.jsps.go.jp/english/]. The funders had no role in study design, data collection and analysis, decision to publish, or preparation of the manuscript.”

4. We note that your Data Availability Statement is currently as follows: “All relevant data are within the manuscript and in Supporting Information files.” Please confirm at this time whether or not your submission contains all raw data required to replicate the results of your study. Authors must share the “minimal data set” for their submission. PLOS defines the minimal data set to consist of the data required to replicate all study findings reported in the article, as well as related metadata and methods.

RESPONSE: We have added a dataset to Supplementary Information as ‘S1 Data. dataset.csv.’ This dataset contains all raw data required to replicate all study findings reported in the revised article.

RESPONSE: As per the response to Comment 4 of the Journal Requirements, we have added a dataset to Supplementary Information as ‘S1 Data. dataset.csv.’ Our entire dataset is now freely accessible.

RESPONSE: We have included our full name of ethics committee and added the approval number in the ‘Materials and methods’ section of our manuscript file. Written informed consent was obtained from all participants before their participation. We have revised the manuscript as follows:

Lines 73–76: “Written informed consent was obtained from all participants before their participation. This study was approved by our institutional ethics review board (Ethics Committee for Epidemiology of Hiroshima University, approval number: E2018-1517) and conducted in accordance with the Declaration of Helsinki.”

RESPONSE: We have ensured that our manuscript contains no retracted references. In addition, we have added following several references to the reference list in order to response to the Reviewers’ comments.

Additional references:

25. Cohen J. Statistical Power Analysis for the Behavioral Sciences. 2nd ed. Lawrence Erlbaum Associates; 1988.33.

33. Bland JM, Altman DG. Statistical methods for assessing agreement between two methods of clinical measurement. Lancet. 1986;1(8476):307-310. doi: 10.1016/S0140-6736(86)90837-835

34. Iizuka T, Tomita Y. Reliability of motion phase identification for long-track speed skating using inertial measurement units. PeerJ. 2024;12:e18102. doi: 10.7717/peerj.18102.41

35. Eto T, Kitamura S, Shikano A, Tanabe K, Higuchi S, Noi S. Estimating dim light melatonin onset time in children using delta changes in melatonin. Sleep Biol Rhythms. 2024;22(2):239-46. doi: 10.1007/s41105-023-00493-x.43

36. Anvari A, Halpern EF, Samir AE. Essentials of Statistical Methods for Assessing Reliability and Agreement in Quantitative Imaging. Acad Radiol. 2018;25(3):391-396. doi: 10.1016/j.acra.2017.09.010.

41. Aguiar LT, Martins JC, Lara EM, Albuquerque JA, Teixeira-Salmela LF, Faria CD. Dynamometry for the measurement of grip, pinch, and trunk muscles strength in subjects with subacute stroke: reliability and different number of trials. Braz J Phys Ther. 2016;20(5):395-404. doi: 10.1590/bjpt-rbf.2014.0173.

42. Hock N, Lindstrom D. Normative data for the Baseline(R) 5 position hydraulic pinch meter and the relationship between lateral pinch strength and pinch span. J Hand Ther. 2021;34(3):453-62. doi: 10.1016/j.jht.2020.03.007.

43. Dempsey PG, Ayoub MM. The influence of gender, grasp type, pinch width and wrist position on sustained pinch strength. International Journal of Industrial Ergonomics. 1996;17(3):259-73. doi: Doi 10.1016/0169-8141(94)00108-1.

44. Shih YC, Ou YC. Influences of span and wrist posture on peak chuck pinch strength and time needed to reach peak strength. International Journal of Industrial Ergonomics. 2005;35(6):527-36. doi: 10.1016/j.ergon.2004.12.002.

45. Heffernan C, Freivalds A. Optimum pinch grips in the handling of dies. Appl Ergon. 2000;31(4):409-14. doi: 10.1016/s0003-6870(99)00064-2.

Additional Editor comments:

The review process for your manuscript titled "Ulnar finger posture effect on a pinch strength" (Manuscript ID: [PONE-D-24-49287]) has been completed. The reviewers have suggested minor revisions to enhance the clarity and quality of your work.

Please find the reviewer's comments for your reference. Kindly revise the manuscript accordingly and submit the revised version along with a point-by-point response to the comments at your earliest convenience.

RESPONSE: We appreciate your revision of our manuscript and the opportunity to resubmit. We successfully incorporated the Reviewers’ feedback into our revised manuscript as discussed below.

Review Comments to the Author

Reviewer #1:

Thank you for the opportunity to review this interesting paper on finger postures during pinching. The experimental objectives, methods, and postoperative data evaluation methods including inter-rater evaluations are all well-documented. There are some questions regarding the discussion of the results. Please refer to the points below.

RESPONSE: We thank the Reviewer for these kind comments. We have revised our manuscript based on the Reviewer’s comments as described below.

COMMENT 1: Line 212, page16; The ICC values for inter-rater reliability (0.52-0.68) are not particularly high; please include a discussion addressing this point. Could this be due to a lack of experience on the part of Examiner 2?

RESPONSE: As the Reviewer pointed out, the lower ICC values for inter-rater reliability compared to intra-rater reliability (0.71–0.92) and test-retest reliability (0.71–0.91) could be due to difference in experience between Examiner 1 and Examiner 2, or to the shorter measurement training time (30 min) for Examiner 2. Our interpretation of these results is that it may be necessary for inexperienced examiners to be provided with sufficient measurement training (e.g., training time, measuring method instructions). We have added this in the ‘Discussion’ section as follows:

Lines 252–259: “On the other hand, the ICC values for inter-rater reliability (0.52–0.68) were lower than those for intra-rater (0.71–0.92) and test-retest (0.71–0.91) reliability. These lower ICC values for inter-rater reliability could be attributed to the difference in experience between Examiner 1 and Examiner 2, or to the insufficient measurement training time (30 min) for Examiner 2. It appears necessary for inexperienced examiners to receive more comprehensive measurement training, such as sessions longer than 30 min or instruction by a supervisor following the ASHT recommended measurement methods [1]”.

COMMENT 2: What are the causes of the differences in the natural ulnar finger posture during pinching? If pinch strength increases when the ring and little fingers are flexed posture, why do some individuals extend their ring and little fingers instead under power pulp pinch?

RESPONSE: People can develop various hand and finger posture during the growth stage, and ulnar finger extension in a pinch movement emerges as part of this process (Hohlstein, J Occup Ther, 1982). A previous study has suggested that the extension of the ulnar finger causes tension in the soft tissues around the index finger, leading to joint stability (Agee et al., J Hand Surg Am, 1980). However, the reason why some individuals extend the ulnar finger under power pulp pinch remains unclear. In future studies, we intend to examine this phenomenon from a biomechanical perspective to clarify why certain individuals extend the ulnar finger during pinching tasks. We have added the future prospects in the ‘Discussion’ section as follows:

Lines 285–289: “A detailed investigation using movement analysis and EMG would be required to clarify the exact influence of ulnar finger posture during pinching movements. These biomechanical and physiological measurements could also provide insight into why the ulnar finger is extended during pinching movements.”

Reviewer #2:

This manuscript is an important finding for hand therapy and outcome measurement assessment of the hand. I believe this work adds to the body of literature with new knowledge but it is not without its limitations. Please see my comments below.

RESPONSE: We thank the Reviewer for these kind comments. We have revised our manuscript based on the Reviewer’s comments, as described below.

COMMENT 1: Can you elaborate on why you chose to focus solely on pulp-to-pulp pinch strength rather than including other pinch types (e.g., lateral or tripod pinch) in this study? How might the inclusion of other pinch types alter the findings?

RESPONSE: As the Reviewer pointed out, this study focused on only pulp-to-pulp pinching and did not include measurements of the tripod pinching. This limitation was intentional to minimize participant fatigue and to accurately isolate the influence of ulnar finger posture during pinching tasks. As suggested by the Reviewer, future studies should measure the strength of tripod pinching measurements with appropriate experimental setup and additional conditions. Although the ulnar finger posture may affect lateral pinch strength, it is clinically uncommon for patients to perform the lateral pinch with the ulnar finger extended. Consequently, we did not compare the lateral pinching between flexion and extension of the ulnar fingers. We have revised the manuscript in the limitation paragraph of the ‘Discussion’ section as follows:

Lines 277–282: “Second, we only measured pulp-to-pulp pinching movements using the thumb and index finger. However, tripod pinching using the thumb, index, and middle fingers may also be influenced by the posture of the ring and little fingers. Future studies should compare the strength of the tripod pinch between flexion and extension of the ulnar fingers while ensuring experimental settings that minimize confounding factors such as examiner variability and fatigue.”

COMMENT 2: The abstract mentions no consensus on ulnar finger posture during pinch strength testing. Could you provide more context on the range of practices currently observed in clinical settings beyond the ASHT recommendation?

RESPONSE: We have added in the ‘Abstract’ that the ulnar finger differs in flexion and extension depending on the measurement method used in the previous study. In addition, we have also revised the ‘Introduction’ session of our manuscript as follows:

Abstract:

Lines 18–21: “…; however, there is no consensus regarding the posture of the three ulnar fingers during pinch strength testing, and existing studies have different flexion or extension posture, depending on the measurement method used”.

Introduction:

Lines 52–54: “…, and depending on the measurement method used, the posture of the ulnar fingers differs in flexion or extension [7, 15-19].”

COMMENT 3: How do you envision these findings being integrated into standardized clinical protocols for pinch strength assessment, given the variability in current practices?

RESPONSE: Since our findings enhance the ASSH’s recommended measurement methods, clinicians should follow a standardized clinical protocol that is integrated into this recommendation. Furthermore, based on the results of this study, which showed moderate ICC for the inter-rater reliability, the same examiner should perform the pinch testing when longitudinally assessing a patient’s pinch strength. We have created a new paragraph in the ‘Discussion’ section to highlight the clinical relevance as follows:

Lines 260–267: “Our findings support the results of a previous study on the effect of ulnar finger posture during the pinching movement [12] and could be integrated into standard protocol of the ASSH's recommended measurement methods [1]. Based on our findings, clinicians should consistently instruct patients to flex their ulnar fingers during pinch testing as the flexion posture generates greater pinch strength and exhibits less inter-, intra-rater, and test-retest variability than the extension posture. Furthermore, given the moderate ICC for the inter-rater reliability, having the same examiner perform pinch testing when longitudinally assessing a patient’s pinch strength is advisable.”

COMMENT 4: You note that the effect of ulnar finger posture on pinch strength is not well explored. Are there specific biomechanical or physiological findings/issues from prior studies that guided your research question?

RESPONSE: We are guided in our research questions on the effect of ulnar finger posture by biomechanical findings. We have added our consideration of the biomechanical findings in the ‘Introduction’ section as fol

---

## [Decision Letter · Decision Letter 1]

13 May 2025

Ulnar finger posture effect on a pinch strength

PONE-D-24-49287R1

Dear Dr Shota Date,

We’re pleased to inform you that your manuscript has been judged scientifically suitable for publication and will be formally accepted for publication once it meets all outstanding technical requirements.

Kind regards,

Saumya Srivastava, Ph.D.

Academic Editor

PLOS ONE

Additional Editor Comments (optional):

Dear Shota Date ,

I am pleased to inform you that the reviewers have completed their evaluation of your manuscript titled "Ulnar finger posture effect on a pinch strength" (PONE-D-24-49287R1). After reviewing their feedback and your revisions, we are happy to confirm that your manuscript has been accepted for publication.

Sincerely,

Dr Saumya Srivastava,

Academic Editor

PLOS ONE

**Comments to the Author**

Reviewer #1: All comments have been addressed

Reviewer #2: All comments have been addressed

2. Is the manuscript technically sound, and do the data support the conclusions?

Reviewer #1: Yes

Reviewer #2: Yes

3. Has the statistical analysis been performed appropriately and rigorously?

Reviewer #1: Yes

Reviewer #2: Yes

4. Have the authors made all data underlying the findings in their manuscript fully available?

Reviewer #1: Yes

Reviewer #2: Yes

5. Is the manuscript presented in an intelligible fashion and written in standard English?

Reviewer #1: Yes

Reviewer #2: Yes

Reviewer #1: (No Response)

Reviewer #2: (No Response)

**Do you want your identity to be public for this peer review?** For information about this choice, including consent withdrawal, please see our Privacy Policy

Reviewer #1: No

Reviewer #2: No

---

## [Editor Report · Acceptance letter]

PONE-D-24-49287R1

PLOS ONE

Dear Dr. Date,

I'm pleased to inform you that your manuscript has been deemed suitable for publication in PLOS ONE. Congratulations! Your manuscript is now being handed over to our production team.

Kind regards,

on behalf of

Dr Saumya Srivastava

Academic Editor

PLOS ONE